# IL6 and IL6R as Prognostic Biomarkers in Colorectal Cancer

**DOI:** 10.3390/biom14121629

**Published:** 2024-12-19

**Authors:** Kathryn A. F. Pennel, Ahmad Kurniawan, Sara Samir Foad Al-Badran, Leonor Schubert Santana, Jean Quinn, Colin Nixon, Phimmada Hatthakarnkul, Noori Maka, Campbell Roxburgh, Donald McMillan, Joanne Edwards

**Affiliations:** 1School of Cancer Sciences, Wolfson Wohl Cancer Research Institute, University of Glasgow, Estate, Glasgow G61 1BD, UKs.al-badran.1@research.gla.ac.uk (S.S.F.A.-B.); leonorpatricia.schubertsantana@glasgow.ac.uk (L.S.S.);; 2Cancer Research UK Scotland Institute, Garscube Estate, Glasgow G61 1BD, UK; c.nixon@crukscotlandinstitute.ac.uk; 3Department of Pathology, Queen Elizabeth University Hospital, Glasgow G51 4TF, UK; 4Academic Unit of Surgery, Glasgow Royal Infirmary, Glasgow G4 0SF, UK

**Keywords:** colorectal cancer, interleukin-6, interleukin-6 receptor, prognosis, biomarker, sidedness, inflammation

## Abstract

Colorectal cancer is the third most diagnosed malignancy worldwide and survival outcomes remain poor. Research is focused on the identification of novel prognostic and predictive biomarkers to improve clinical practice. There is robust evidence in the literature that inflammatory cytokine interleukin-6 (IL6) is elevated systemically in CRC patients and that this phenomenon is a predictor of poor survival outcome. However, evidence is more limited for the role of IL6 and its cognate receptor, IL6R, within the tumour epithelium and microenvironment. This study aimed to investigate IL6 and IL6R expression in a large cohort of retrospectively collected patient tumour specimens and determine association with clinical outcomes and characteristics. High expression of IL6R in the tumour epithelium was associated with reduced cancer-specific survival in patients with right-sided colon cancer. In these patients, high IL6R expression was also associated with an increased systemic neutrophil-to-lymphocyte ratio. A high number of copies of *IL6* mRNA within the tumour-associated stroma, but not epithelium, was associated with reduced cancer-specific survival. The results from this study have validated IL6R as a marker of poor prognosis in a subgroup of CRC patients and identified the spatially resolved prognostic nature of intra-tumoural *IL6* expression. This study has also highlighted the need for investigation of IL6/IL6R-targeted therapies as novel treatment strategies for patients with colon cancer.

## 1. Introduction

Colorectal cancer (CRC) is a leading cause of worldwide cancer-related mortality. Existing therapeutics for CRC patients include surgical resection and adjuvant or neoadjuvant chemotherapy/chemoradiotherapy, mainly informed through pathological staging (tumour node metastasis (TNM)) and the tumour subsite. Survival outcomes remain suboptimal, and thus research is focused on improving methods for predicting response and prognosis, and understanding of mechanisms underlying tumourigenesis to identify novel therapeutic targets.

Systemic inflammation is a well-established phenomenon associated with disease progression [1,2]. Interleukin-6 (IL6) is an inflammation-associated cytokine known to be predictive of poor disease-specific outcomes when upregulated in the serum of patients with solid tumours such as non-small cell lung cancer, prostate cancer and CRC [2,3,4,5]. High systemic IL6 levels are associated with adverse clinicopathological characteristics in CRC, including advanced T stage and tumour necrosis [2].

Evidence for the role of IL6 and its cognate receptor (IL6R) within the tumour itself and surrounding microenvironment remains a key area of research. IL6R can be expressed within tumour cell membranes or can exist in a soluble state within the tumour cell cytoplasm. In the context of cancer, IL6 can be produced by tumour cells and other populations in the tumour microenvironment (TME), including macrophages and inflammatory cancer-associated fibroblasts [6,7]. Upon IL6 binding to IL6R, a signal is transduced through gp130, which results in downstream activation of various transcription factors, including signal transducer and activator of transcription 3 (STAT3), nuclear factor kappa-light chain enhancer of activated B cells (NFκB) and mitogen-activated protein kinase (MAPK) [8].

A previous retrospective study of breast cancer reported that high expression of IL6R within tumour cells was associated with reduced cancer-specific survival (*n* = 593) [9]. Conversely, in lung adenocarcinoma (*n* = 140), high IL6R protein expression predicted improved survival outcomes, and expression was higher in adjacent normal tissue compared to tumour epithelium [10]. In colon cancer, high levels of *IL6* at the bulk mRNA level have been observed to predict increased risk of relapse [11]. However, more recently there has been emerging evidence that the presence of a high number of IL6+ immune cells at the invasive front of the tumour is predicative of better overall survival in CRC [12]. Data from in vitro models implicates macrophage-derived IL6 in CRC cells acquiring resistance to chemotherapy treatment [7]. These studies highlight the context-dependant and pleiotropic roles of IL6/IL6R in the TME, and the need for further research in the field.

This study aimed to investigate the expression of IL6R within a large cohort of retrospectively collected CRC patient biospecimens via immunohistochemistry and IL6 via RNA sequencing and RNAScope^®^ to determine association with patient prognosis and clinicopathological characteristics.

## 2. Methods

### 2.1. Patient Cohorts

#### 2.1.1. Cohort 1

To assess the prognostic value of *IL6* and IL6R in CRC, a large retrospectively collected patient cohort of stage II-IV CRC tumours was utilised (*n* = 1030). Samples were previously incorporated into a tissue microarray (TMA), which consisted of 0.6 mm cores in triplicate for each patient to account for tumour heterogeneity. Tumours were removed via surgical resection with curative intent within the National Health Service Greater Glasgow and Clyde health board between 1997 and 2007. Tumours were staged using the 5th edition of Tumour Node Metastasis (TNM) staging [13]. After exclusion of cases due to mortality within 30 days of surgery or administration of neoadjuvant therapy, *n* = 915 patients were included in downstream analysis. At the time of last follow-up, *n* = 339 (37%) patients were alive, *n* = 280 (30.6%) had died of cancer and *n* = 296 (32.3%) had died of other causes. The mean follow-up time was 89 months. Ethical approval was in place prior to commencement of the project (16/WS/0207).

#### 2.1.2. Cohort 2

A large retrospectively collected patient cohort of stage II-III CRC patient tissue was utilised as a validation cohort for this study *(n* = 787). Samples were previously incorporated into a tissue microarray (TMA), which consisted of 0.6 mm cores in triplicate for each patient, in concordance with cohort 1. Tumours were removed via surgical resection with curative intent within the National Health Service Greater Glasgow and Clyde health board between 1997 and 2013. Tumours were staged using the 5th edition of TNM staging. After exclusion of patients who died within 30 days of surgery and/or those who received neoadjuvant therapy, *n* = 715 cases remained. At the time of last follow-up, *n* = 247 (34.6%) patients were alive, *n* = 205 (28.7%) had died of cancer and *n* = 262 (36.7%) had died of causes unrelated to their primary cancer. The median follow-up time was 93.48 months. Ethical approval was in place for the study (MREC/01/0/36).

### 2.2. Immunohistochemical Staining

TMAs were stained for IL6R via immunohistochemistry (IHC). Sections were dewaxed in Histoclear (National Diagnostics, Charlotte, NC, USA) and rehydrated through a series of graded alcohols. Antigen retrieval was performed by heating slides under pressure in citrate buffer pH6. Endogenous peroxidases were blocked using 3% H_2_O_2_ before blocking in 10% casein. Sections were incubated in primary antibody (IL6R, #128008, Abcam, Cambridge, UK) diluted 1:3000 in antibody diluent (S0809, Agilent Technologies, Santa Clara, CA, USA) overnight at 4 °C. Tissue was washed in tris-buffered saline (TBS) solution and incubated in ImmPRESS secondary antibody (Vector Laboratories, Newark, CA, USA) at room temperature. Sections were washed in TBS and DAB-chromogen applied for 5 min before counterstaining, dehydration and exposure to Histoclear (National Diagnostics, Charlotte, NC, USA). Sections were mounted using Omnimount (National Diagnostics, Charlotte, NC, USA) and scanned onto an NZ connect digital viewing platform using a Hamamatsu NanoZoomer (Hammamatsu, Shizuoka, Japan).

### 2.3. Histological Scoring

Staining intensity was scored via weighted histoscore by a single observer (cohort 1-KP, cohort 2-AK). Scores were validated through attainment of an intraclass correlation coefficient of >0.7 in 10% of the cohort using independent scores from a second observer (cohort 1-SAB, cohort 2-KP). Cohort 1 was scored manually and cohort 2 was scored using QuPath [14]. This was performed as previously described [15]. The continuous scores ranging from 0–300 were dichotomised based on cut points for high and low expression, determined using Survminer version 0.4.9 in R Studio version 3.6.0 based on cancer-specific survival.

### 2.4. RNAScope^®^

TMAs from cohort 1 were probed for *IL6* RNA by in situ hybridisation using RNAScope^®^ (ACD Bio, Newark, CA, USA). Staining was performed using a Leica Bond Rx system (Leica Biosystems, Wetzlar, Germany) at the CRUK Scotland Institute (CN). For quantification, a classifier was built to distinguish between tumour epithelium and stromal areas using Halo software version 3.6.4134.396 (Indica Labs, Albuquerque, NM, USA). Prescence of *IL6* was semi-quantitively assessed through copies per µM^2^ normalised to the presence of Peptidylprolyl Isomerase B (*PPIB*) housekeeping control. Cut offs for high and low expression were determined using Survminer version 0.4.9 in R Studio (RStudio, Boston, MA, USA) with R version 3.6.0 based on cancer-specific survival.

### 2.5. TempOSeq RNA Sequencing

Full transcriptome RNA sequencing was performed on archival tissue from patient cohort 2 by TempOSeq profiling (BioClavis, Glasgow, UK) as previously described [16]. Data analysis was performed using R studio version 2024.04.2+764 and R version 4.3.3. Batch correction was performed using the sva package version 3.50.0 and quantile normalisation, and gene filtering was performed using the edgeR package version 4.0.16. Normalised counts for IL6 were extracted and a cut point for high and low expression was determined using Survminer version 0.4.9 in R Studio (RStudio, Boston, MA, USA) with R version 3.6.0 based on cancer-specific survival.

### 2.6. Statistical Analysis

Analysis of data was performed by generating Kaplan Meier curves using Survival version 3.7-0 in R Studio with R version 3.6.0 (RStudio, Boston, MA, USA). Univariate cox regression and chi-squared tables of association were performed and generated in SPSS version 29 (IBM SPSS, Chicago, IL, USA). Box plots were generated in GraphPad Prism version 9 (GraphPad Software, La Jolla, CA, USA).

## 3. Results

### 3.1. High Expression of Interleukin-6 Receptor Predicts Poor Clinical Outcomes in Right-Sided Colon Cancer

Immunohistochemical staining of CRC tissue specimens from cohort 1 yielded a range of staining intensities for IL6R in the tumour cell cytoplasm, as shown by representative images (Figure 1A). When staining was semi-quantitively measured, weighted histoscores ranged from 0 to 200. Continuous scores were dichotomised into high and low expression groups based on a cut point of 44.9 identified using the Survminer package. This resulted in *n =* 504 (87.5%) patients classified as low and *n* = 72 (12.5%) patients classified as high for cytoplasmic IL6R expression. Kaplan Meier survival analysis showed that in the full cohort, high IL6R expression trended towards an association with reduced cancer-specific survival (CSS) (*HR* = 1.472, 95%CI; 0.948–2.284, *p* = 0.082) (Figure 1B). These results were validated in a second independent patient cohort. High expression of IL6R in the tumour cell cytoplasm of cohort 2 also trended towards an association with reduced CSS in the full cohort (*HR* = 1.469, 95%CI; 0.968–2.228, *p* = 0.068) (Figure 1C).

Given the evidence in the literature for differences in the underlying biology of tumours arising in different subsites of the bowel, we investigated if expression of IL6R was different between right-sided or left-sided colon and rectal cases. Despite no difference in expression intensity between subsites (Figure 2A), the association between high IL6R and reduced CSS was potentiated in right-sided colon cases in cohort 1 (*HR* = 2.402, 95% CI; 1.117–5.164, *p* = 0.020) (Figure 2B). There was no association between IL6R expression and outcome in left-sided colon or rectal cancers (Appendix A). In concordance with cohort 1, no difference in IL6R expression intensity was observed between subsites in cohort 2 (Figure 2C), and the association between high IL6R expression and poorer outcome was potentiated in right-sided colon cases (*HR* = 1.995, 95%CI; 1.083–3.675, *p* = 0.024) (Figure 2D). Similarly, there was no association in left-sided colon or rectal cancers (Appendix A).

### 3.2. High Expression of Interleukin-6 Receptor Associates with Adverse Clinicopathological Characteristics in Right-Sided Colon Cancers

To determine association between IL6R expression and clinical characteristics, chi-squared tests were performed in patients with right-sided colon cancer from both cohorts. In patient cohort 1, we identified a significant association between high IL6R expression and presence of marginal involvement (*p* = 0.038) and high Peterson index (*p* = 0.010) (Table 1). In patient cohort 2, high IL6R expression was significantly associated with presence of venous invasion (*p* < 0.001) (Table 2). Kruskal Wallis tests showed patients with high IL6R expression had significantly higher systemic neutrophil-to-lymphocyte ratio in both cohort 1 (*p* = 0.028) and cohort 2 (*p* = 0.049) within right-sided colon cases (Appendix A).

Next, we aimed to explore underlying differences in right-sided tumours with high IL6R expression between patients who had died of cancer versus those still alive/non-cancer-associated mortalities. Across both patient cohorts, the high IL6R expression, poor outcome patients had significantly higher numbers of lymph node metastases (LN met) (cohort 1; *p =* 0.003, cohort 2; *p* = 0.035). Interestingly, Kaplan Meier survival analysis showed within right-sided cases, IL6R expression was only prognostic in the cases with LN mets detected (cohort 1; *p* = 0.016, cohort 2; *p* = 0.031), and not in patients with no LN mets (cohort 1; *p* = 0.42, cohort 2; *p* = 0.79). (Appendix A).

### 3.3. The Prognostic Value of Interleukin-6 Is Spatially Resolved

IL6 is the cognate binding partner for IL6R; therefore, we investigated its expression within bulk transcriptomics data available from a subset of patient cohort 2 (*n* = 319). High expression of *IL6* was associated with reduced cancer-specific survival (*p* = 0.025) (Figure 3A). Although this result was interesting, bulk RNA sequencing did not permit analysis of *IL6* with respect to different tumour compartments. Therefore, we next sought to explore expression of *IL6* within the tumour epithelium and stromal areas of TMAs from patient cohort 1. Due to the secretory nature of IL6, IHC was not an appropriate technique. During optimisation, IHC staining for IL6 yielded a brown blush staining pattern observed across the whole tissue with no specificity to cell type or location, and therefore RNAScope^®^ was utilised.

Positive copies of *IL6* mRNA were detected in both tumour epithelium and the surrounding stroma. Representative images of tissue from cohort 1 probed for *IL6* and housekeeping gene *PPIB* are shown in Figure 3B. *IL6* was detected at higher levels in the tumour-associated stroma versus the tumour epithelial regions (*p* < 0.00001) (Figure 3C), and this could be validated through analysis of an independent RNASeq-based CRC dataset GSE35602 using confoundR (https://confoundr.qub.ac.uk/ accessed on 17 September 2024) (*p* < 0.001) (Figure 3D). Stromal *IL6* showed a moderate correlation with epithelial *IL6* expression (*Rho* = 0.487, *p* < 0.001) (Figure 3E).

Kaplan Meier survival analysis indicated that high expression of *IL6* within the tumour-associated stroma was predictive of reduced CSS in the full cohort (*HR* = 1.974, 95%CI; 1.009–3.860, *p* = 0.042) (Figure 4A). Conversely, there was no association between CSS and *IL6* quantified within the tumour epithelium (*HR* = 0.997, 95%CI; 0.701–1.418, *p* = 0.987) (Figure 4B). There was no difference between tumoural or stromal *IL6* expression relative to tumour sidedness (Appendix A), and contrary to IL6R, there was no association between *IL6* in either compartment with CSS when patients were further stratified into right-sided colon cases (Appendix A).

## 4. Discussion

These data have highlighted a prognostic role for IL6R expression within the tumour cell cytoplasm of right-sided colon cancers. Across two large independent retrospective cohorts, high cytoplasmic IL6R was associated with reduced CSS in patients with right-sided disease. Previous data have highlighted an association between right-sided colonic tumours and elevated systemic inflammation [17]. Other studies have shown that patients with right-sided disease exhibit worse clinical outcomes and present at a more advanced disease stage [18,19]. This highlights the importance of research into the underlying biology of right-sided tumours, and the unmet clinical need for the development of new treatment strategies for this patient subgroup. Despite evidence for IL6R as a marker of poor prognosis from other solid tumour types, including breast and high-grade ovarian [9,20], there are currently limited data in the literature for a prognostic role of IL6R in CRC. Therefore, the findings from the present study represent an important step forward. Although REMARK guidelines have been adhered to through utilisation of a discovery and validation array, future work should involve validation of these findings in a further independent geographically distinct cohort. A further limitation includes the unequal distribution of right-sided, left-sided and rectal cases within the cohorts utilised; however, analysis of IL6R in additional patient cohorts to confirm potentiation in right-sided disease was biologically relevant, and not due to higher power from more cases than the other tumour subsites.

In the present study, we have also shown that the spatial distribution of *IL6* is important in tumour progression. There was a significant association between high *IL6* expression in the stromal regions and reduced cancer-specific survival; however, no relationship between *IL6* located within the tumour and outcome was identified. This was an unexpected finding, and it could be postulated that this is a result of the cellular source of IL6 production being a key factor. A similar study in *n* = 209 CRC patients showed results concordant with this study, implicating high levels of IL6 within the tumour-associated stroma with worse outcomes [21]. The authors also reported a significant association between increased stromal IL6 and reduced influx of anti-tumour CD3+ and CD4+ T cells, and mass cytometry analysis indicted an increased presence of immunosuppressive populations in the TME of the high IL6 subgroup [21]. Similarly, data from another study showed high IL6 expression measured across the entire tumour and TME associated with reduced disease-free survival, and this was potentiated in patients with high PDL1-expressing tumours (*n* = 108) [22]. However, another recent study showed that high expression of IL6+ immune cells within the stroma and invasive front of CRC tumours was associated with favourable clinical characteristics and better survival outcomes in early-stage disease (*n* = 153) [12]. The cohort utilised in the present study consisted of a larger number of cases than previous studies (*n* = 1030), which represents a strength of this work; however, RNAScope^®^ should be repeated in a subsequent independent cohort for validation purposes.

Future work should also include performing multiplex immunofluorescent staining for immune cell and fibroblast markers alongside IL6 to investigate if the prognostic role of stromal IL6 is driven by the presence of specific cell populations in the stroma. A limitation of this study was the use of RNAScope^®^ to detect *IL6* given that mRNA may not be translated to protein. Several other studies have successfully employed IHC to detect IL6 at the protein level; however, we were unable to identify an antibody with appropriate specificity for this study. Unlike RNAScope^®^, IHC is already routinely performed in diagnostic laboratories and, therefore, if IL6 could be accurately detected by IHC this would aid the clinical translation y as a prognostic marker.

## 5. Conclusions

This study has highlighted a novel and significant association between high expression of IL6R and worse clinical outcomes in right-sided colon cancer patients. Similarly, high expression of stromal *IL6* at the mRNA level acted as a marker of poorer cancer-specific survival in CRC.

## Figures and Tables

**Figure 1 biomolecules-14-01629-f001:**
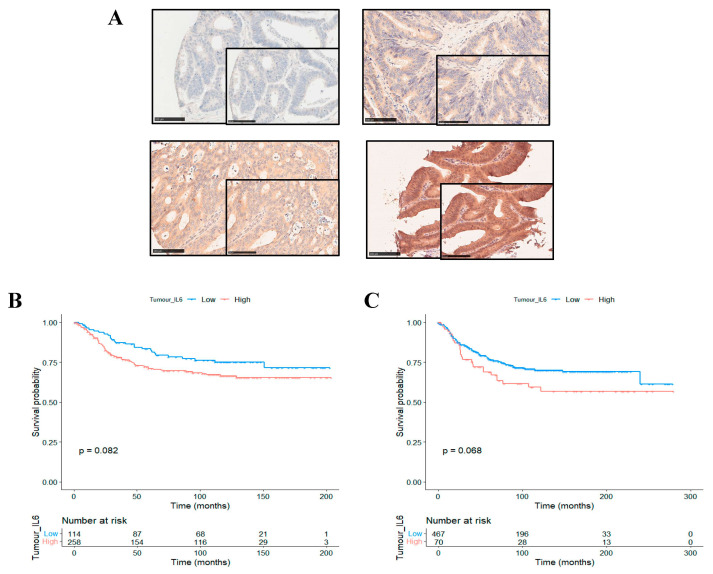
IL6R as a prognostic marker in colorectal cancer. Representative images showing a range of intensities of immunohistochemical staining for IL6R in colorectal cancer patient tissue (**A**). Kaplan Meier curve showing the association between cytoplasmic tumoural IL6R expression and cancer-specific survival in patient cohort 1 (**B**) and cohort 2 (**C**).

**Figure 2 biomolecules-14-01629-f002:**
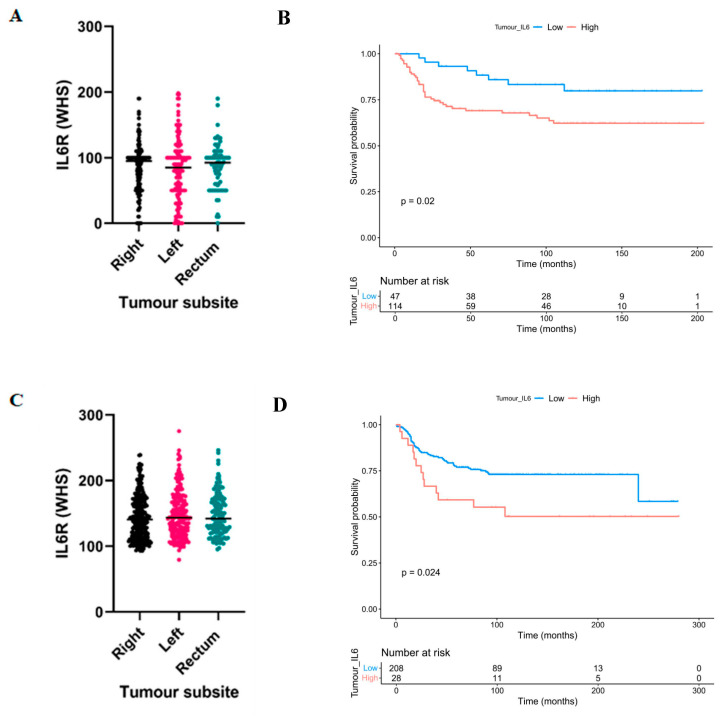
IL6R as a prognostic marker in right-sided colon cancer. Box plot showing the distribution of weighted histoscores of cytoplasmic tumour IL6R across right-sided, left-sided and rectal cancers in patient cohort 1 (**A**). Kaplan Meier curve showing the association between cytoplasmic tumoural IL6R expression and cancer-specific survival in right-sided colon cancer cases from patient cohort 1 (**B**). Box plot showing the distribution of weighted histoscores of cytoplasmic tumour IL6R across right-sided, left-sided and rectal cancers in patient cohort 2 (**C**). Kaplan Meier curve showing the association between cytoplasmic tumoural IL6R expression and cancer-specific survival in right-sided colon cancer cases from patient cohort 2 (**D**).

**Figure 3 biomolecules-14-01629-f003:**
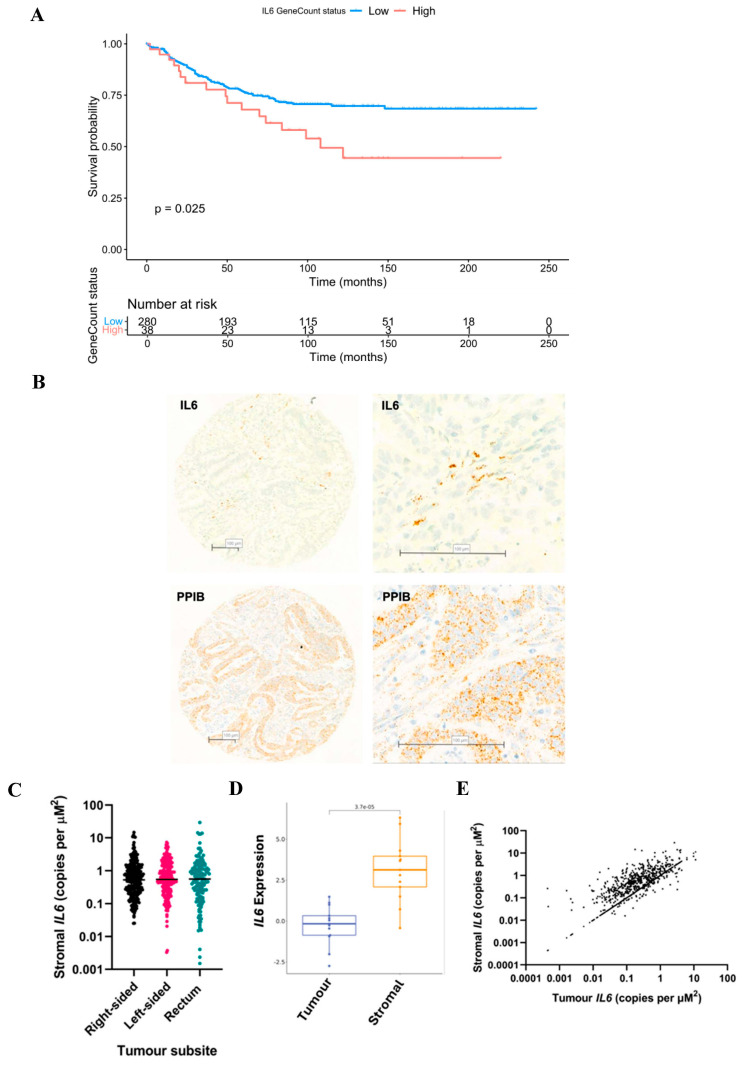
IL6 can be detected in the tumour epithelium and tumour-associated stroma of colorectal cancer patient specimens. Kaplan Meier curve showing the association between IL6 gene expression from RNA sequencing (*n* = 319) and cancer-specific survival in cohort 2 (**A**). Representative images showing colorectal cancer tissue from patient cohort 1 probed for IL6 mRNA via RNAScope^®^ (**B**). Housekeeping control gene peptidyl-prolyl cis-trans isomerase B (PPIB) also shown (**B**). Box plot showing the quantification of IL6 in stromal and tumour compartments of patient cohort 1 analysed by the Mann–Whitney test (**C**). Box plot showing the quantification of IL6 in stromal and tumour compartments of a publicly available dataset using confoundR analysed by the Mann–Whitney test (**D**). Scatter plot for visualisation of the correlation between tumour and stromal IL6 in patient cohort 1 (**E**).

**Figure 4 biomolecules-14-01629-f004:**
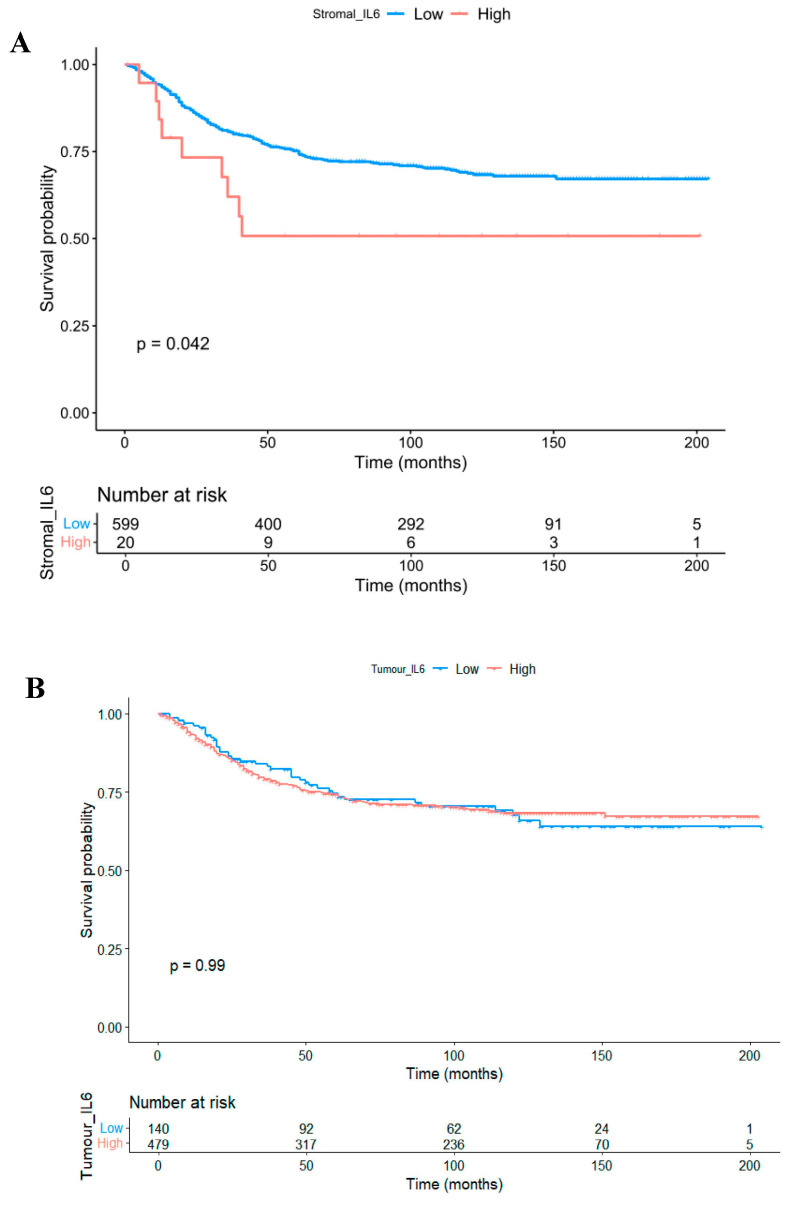
IL6 as a prognostic marker in colorectal cancer. Kaplan Meier curves showing the association between stromal IL6 expression (**A**) and intra-tumoural IL6 expression (**B**) and cancer-specific survival in patient cohort 1.

**Table 1 biomolecules-14-01629-t001:** Association between IL6R and clinicopathological characteristics in cohort 1. Chi-squared table of association. Bold text represents a significant *p* value of α < 0.05.

Clinical Characteristic	Low IL6R	High IL6R	*p*
**Age**			0.324
<65	15 (31.3)	30 (26.3)	
>65	33 (68.8)	84 (73.7)	
**Sex**			0.061
Female	29 (60.4)	52 (45.6)	
Male	19 (39.6)	62 (54.4)	
**T stage**			0.602
1	1 (2.1)	2 (1.8)	
2	5 (10.4)	9 (7.9)	
3	31 (64.6)	65 (57.0)	
4	11 (22.9)	38 (33.3)	
**N stage**			0.298
0	34 (70.8)	67 (58.8)	
1	10 (20.8)	30 (26.3)	
2	4 (8.3)	17 (14.9)	
**mGPS**			0.059
0	20 (60.6)	34 (38.6)	
1	8 (24.2)	25 (28.4)	
2	5 (15.2)	29 (33.0)	
**MMR status**			0.335
pMMR	33 (68.8)	83 (73.5)	
dMMR	15 (31.3)	30 (26.5)	
**Marginal Involvement**			**0.038**
Absent	48 (100.0)	105 (92.1)	
Present	0 (0)	9 (7.9)	
**Peterson Index**			**0.010**
Low	43 (89.6)	82 (71.9)	
High	5 (10.4)	32 (28.1)	
**Vascular Invasion**			0.055
Absent	36 (75.0)	69 (60.5)	
Present	12 (25.0)	45 (39.5)	
**Peritoneal Involvement**			0.179
Absent	37 (77.1)	78 (68.4)	
Present	11 (22.9)	36 (31.6)	
**Tumour perforation**			0.204
Low	47 (97.9)	104 (91.2)	
Moderate	0 (0)	5 (4.4)	
High	1 (2.1)	5 (4.4)	
**GMS**			0.745
0	13 (27.7)	34 (30.1)	
1	27 (57.4)	55 (48.7)	
2	7 (14.9)	24 (21.2)	

**Table 2 biomolecules-14-01629-t002:** Association between IL6R and clinicopathological characteristics in cohort 2. Chi-squared table of association. Bold text represents a significant *p* value of α < 0.05.

Clinical Characteristic	Low IL6R	High IL6R	*p*
**Age**			0.778
<65	63 (30.1)	8 (28.6)	
>65	146 (69.9)	20 (71.4)	
**Sex**			0.460
Female	96 (45.9)	12 (42.9)	
Male	113 (54.1)	16 (57.1)	
**T stage**			0.294
1	5 (2.4)	0 (0)	
2	17 (8.1)	0 (0)	
3	115 (55.0)	18 (64.3)	
4	72 (34.4)	10 (35.7)	
**N stage**			0.163
0	127 (60.8)	11 (39.3)	
1	56 (26.8)	14 (50.0)	
2	26 (12.4)	3 (10.7)	
**mGPS**			0.499
0	112 (53.6)	12 (42.9)	
1	46 (22.0)	9 (32.1)	
2	51 (24.4)	7 (25.0)	
**MMR status**			0.093
dMMR	37 (32.3)	10 (40.0)	
pMMR	129 (77.7)	15 (60.0)	
**Marginal Involvement**			0.254
Absent	197 (94.3)	25 (89.3)	
Present	12 (5.7)	3 (10.7)	
**Peterson Index**			0.521
Low	153 (73.2)	21 (75.0)	
High	56 (26.8)	7 (25.0)	
**Vascular Invasion**			**<0.001**
Absent	80 (38.3)	20 (71.4)	
Present	129 (61.7)	8 (28.6)	
**Peritoneal Involvement**			0.591
Absent	149 (71.3)	20 (71.4)	
Present	60 (28.7)	8 (28.6)	
**Tumour perforation**			0.335
No	201 (96.2)	26 (92.9)	
Yes	8 (3.8)	2 (7.1)	
**GMS**			0.120
0	27 (13.2)	2 (7.4)	
1	141 (69.1)	17 (63.0)	
2	36 (17.6)	8 (29.6)	

## Data Availability

Data are stored and accessible through Glasgow Safehaven (*cohort 1-GSH/18/ON007*), (*cohort 2-GSH21ON009*). RNA sequencing data are available through accession number PRJNA997336.

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
