# Peer review of "IL6 and IL6R as Prognostic Biomarkers in Colorectal Cancer"

_biomolecules, 2024, doi:10.3390/biom14121629_

Round 1
Reviewer 1 Report
Comments and Suggestions for Authors
Pennel et al. performed a study of the relationship between IL6 expression or IL6 receptor expression and the prognosis of colorectal cancer using two large cohorts of retrospectively collected tumor patient specimen. This reviewer found that the data derived from valuable clinical samples support their conclusions.
A few minor points:
1. Font sizes for all the figures depicting Kaplan Meier curves need to be enlarged so that the words can be clearly seen. The time units were not indicated in these figures.
2. Specify how IL6 expression is depicted in Fig. 3D. Unit? Or relative expression?
3. Line 75-76, a reference should be added for the tumor staging.
4. Line 148, Is “IL6R in the tumor cell cytoplasm” a phenomenon for tumor cells, as it is supposed to be a membrane protein in normal tissue? Give explanation or speculation of the reasons for this phenomenon.
5. CSS is the abbreviation for “cancer specific survival”? This needs to be specificized at its first appearance in the manuscript.
Author Response
Comment 1 Font sizes for all the figures depicting Kaplan Meier curves need to be enlarged so that the words can be clearly seen. The time units were not indicated in these figures.
Response
- Thank you for your suggestions, we have amended the font size and added the time units
Comment 2 Specify how IL6 expression is depicted in Fig. 3D. Unit? Or relative expression?
Response
- Thanks for your comments, we have used the expression data from publicly available confoundR which has an expression from single cell dataset
Comment 3 Line 75-76, a reference should be added for the tumor staging.
Response
- Thank you very much for your concern, we have added the reference regarding the tumour staging in line 79
Comment 4 Line 148, Is “IL6R in the tumor cell cytoplasm” a phenomenon for tumor cells, as it is supposed to be a membrane protein in normal tissue? Give explanation or speculation of the reasons for this phenomenon.
Response
- Thank you very much for your insightful comment. IL6R although usually found within the membrane, however, when cleaved by proteases the soluble form can be found in the cytoplasm. It is well documented in the literature as it allows for trans-signaling as opposed to classical signaling observed with membrane bound IL6R.
Comment 5 CSS is the abbreviation for “cancer specific survival”? This needs to be specificized at its first appearance in the manuscript.
Response
- Thank you very much for this, we have specified the word CSS as first appear in line 156-157
Reviewer 2 Report
Comments and Suggestions for Authors
In this article, the authors explore the biomarker capabilities of IL6R and IL6 in colorectal cancer by using their own two cohorts (n1=1030 and n2=787) of patients. The expression is analyzed given anatomical (right-sided, left-sided and rectum), tumour tissue (tumour/stroma) and cellular level (cytoplasm) specificities.
General comment:
The article provides a detailed description of IL6R and IL6 expression in tissues from colorectal cancer patients and their correlation with the cancer-specific survival. The research topic of this article is highly relevant and the data on IL6R and IL6 expression obtained on such large cohorts are valuable. The strengths and limitations of the study are fairly presented and the work performed is methodologically sound. The manuscript is well written, especially the Discussion part which gives a concise overview of the relevant literature and offers biological explainations of the presented results.
Minor points:
*Figure 2 captions: There is no C) and D) caption; however, A) and B) are listed twice. Moreover, the weighted histoscores are quite different for cohort 1 and 2; the authors could briefly explain why.
*Figure 1B and 1D: Although the descriptions 'IL6R=0' and 'IL6R=1' are quite intuitive, it would be better to use more specific description (low/high or similar). The same comment for other Figures where this is applicable.
*Abbreviation CSS (line 156) should be explained, as in lines 226 and 277.
Author Response
Comment 1 *Figure 2 captions: There is no C) and D) caption; however, A) and B) are listed twice. Moreover, the weighted histoscores are quite different for cohort 1 and 2; the authors could briefly explain why.
Response
- Thank you very much for your insightful comments, we have edited the captions highlighted in yellow. Regarding the different in the weighted histoscores, we acknowledged the use of manual scoring and digital scoring which could be a limitation of the study. We have included this in the discussion
Comment 2 *Figure 1B and 1D: Although the descriptions 'IL6R=0' and 'IL6R=1' are quite intuitive, it would be better to use more specific description (low/high or similar). The same comment for other Figures where this is applicable.
Response
- Thank you very much for your suggestion, we have edited all the label from 0 to low and 1 to high in all the survival plots